# Diffusive lensing as a mechanism of intracellular transport and compartmentalization

**Achuthan Raja Venkatesh[1,2], Kathy H Le[1], David M Weld[3]\*, Onn Brandman[1]\***

[1]Department of Biochemistry, Stanford University, Stanford, United States; [2]Department of Biological Sciences, Indian Institute of Science Education and Research (IISER) Mohali, Mohali, India; [3]Department of Physics, University of California, Santa Barbara, Santa Barbara, United States

**Abstract** While inhomogeneous diffusivity has been identified as a ubiquitous feature of the cellular interior, its implications for particle mobility and concentration at different length scales remain largely unexplored. In this work, we use agent-based simulations of diffusion to investigate how heterogeneous diffusivity affects the movement and concentration of diffusing particles. We propose that a nonequilibrium mode of membrane-less compartmentalization arising from the convergence of diffusive trajectories into low-diffusive sinks, which we call 'diffusive lensing,' is relevant for living systems. Our work highlights the phenomenon of diffusive lensing as a potentially key driver of mesoscale dynamics in the cytoplasm, with possible far-reaching implications for biochemical processes.

\*For correspondence:
weld@physics.ucsb.edu (DMW);
onn@stanford.edu (OB)

**Competing interest:** The authors declare that no competing interests exist.

## eLife assessment

The authors discuss an effect, "diffusive lensing", by which particles would accumulate in high-viscosity regions – for instance in the intracellular medium. To obtain these results, the authors rely on agent-based simulations using custom rules performed with the Ito stochastic calculus convention. The "lensing effect" discussed is a direct consequence of the choice of the Ito convention without spurious drift which has been discussed before and its adequacy for the intracellular medium is insufficiently discussed and relatively doubtful. Consequently, the relevance of the presented results for biology remain unclear and based on **incomplete** evidence.

## Introduction

Diffusion is a fundamental phenomenon of transport at scales ranging from atoms to galaxies. In cells, diffusion of individual components occurs in a complex, crowded milieu (*Ellis, 2001*; *Luby-Phelps, 1999*; *van den Berg et al., 2017*) that is known to exhibit position-dependent diffusivity (*Berret, 2016*; *Garner et al., 2023*; *Huang et al., 2022*; *McLaughlin et al., 2020*; *Šmigiel et al., 2022*; *Xiang et al., 2020*). Diffusion can occur within or between cellular compartments, where concentrated components carry out chemical reactions. This rich interaction of diffusion and compartmentalization provides the context for cellular biochemistry. Diffusivity varies inversely with viscosity, a key biophysical parameter of the cytoplasm (*Bausch et al., 1999*; *Hu et al., 2017*) that dictates translational and rotational mobility of proteins and, by extension, possibly influences their activity (*Huang et al., 2022*; *Lippincott-Schwartz et al., 2001*; *Pan et al., 2009*). While diffusivity has been implicated in modulating or driving a range of cellular processes (*Molines et al., 2022*; *Persson et al., 2020*; *Xie et al., 2022*), the role of *inhomogeneous* diffusivity in shaping biochemistry by regulating biomolecular

concentration and dynamics remains poorly understood. Observation of diverse instances of accumulation across scales motivates our search for uncovering how space-dependent diffusivity affects cell biology. The accumulation of small molecules within the nuclear pore, for instance, has been attributed to diffusion through a viscous region (*Ma et al., 2012*). At the macroscale, Chladni patterns are an example of particle concentration resulting from inhomogeneous stochastic transport coefficients (*Grabec, 2017*). The implications of inhomogeneous diffusivity as a nonequilibrium phenomenon occurring at time scales and length scales relevant to biology remain largely unexplored. Theoretically, more information is required to specify the problem than just the diffusion constant: different mathematical interpretations of the stochastic term in diffusion equations with a spatially inhomogeneous diffusion constant result in different physical predictions (see *Appendix* for more information). Interestingly, diverse mesoscale outcomes are also seen in the case of active biological matter (*Bechinger et al., 2016*; *Needleman and Dogic, 2017*; *Yeomans, 2017*), the density-dependent concentration of active Brownian particles (*Cates and Tailleur, 2015*) and size-dependent condensation kinetics in the case of *C. elegans* colony formation (*Chen and Ferrell, 2021*). While these phenomena focus on motile energy-expending tracers, here we emphasize the underlying space-dependency of a physical property characterizing diffusion. In particular, accumulation arising from inhomogeneous diffusivity may represent a novel mechanism of effective compartmentalization, a key activity for cells in regulating biochemical processes.

In this work, we employ agent-based modeling to explore how position-dependent diffusivity can affect the distribution of tracer particles. We show that under a set of assumptions that relate to the ambiguities intrinsic to modeling inhomogeneous diffusivity (see *Appendix*), transport due to a diffusivity gradient leads to particle trajectories being biased toward areas of lower diffusivity, leading to effective compartmentalization and the growth of concentration gradients; we call this effect 'diffusive lensing,' in non-quantitative analogy to the effects on light rays of media with inhomogeneous refractive index, including refraction and the formation of caustics. Analyzing particle trajectories, we show that diffusive lensing manifests differently from homogeneous diffusion at the emergent scale. We conclude that inhomogeneous diffusivity may have diverse implications for intracellular transport, from sequestering particles to modulating where and when higher-order processes such as clustering happen, in a way that is not predictable from equivalent homogeneous-diffusivity models and could affect biochemical reactions.

## Results

### Inhomogeneous diffusivity drives particle accumulation

We probed the effect of inhomogeneous diffusivity on particle concentration using agent-based modeling of particle dynamics (*Figure 1—figure supplement 1A*; see *Methods*). In our modeling, the expected macroscale behavior is dictated by the Itô interpretation of heterogeneous diffusion (see *Appendix*) (*Volpe and Wehr, 2016*). Our model was non-anticipatory in that for a modeled particle traversing an inhomogeneous diffusivity, the step size distribution was defined by the diffusivity at its *present* position. Other equally consistent interpretations (such as the entirely anticipatory 'isothermal' interpretation) produce different macroscale behaviors (*Figure 1—figure supplement 1B*). The range of physically incompatible possibilities resulting from different interpretations is known as the Itô-Stratonovich dilemma (*Lau and Lubensky, 2007*; *Sokolov, 2010*; *Tupper and Yang, 2012*; *Van Kampen, 1988*; *Volpe and Wehr, 2016*). For systems at thermal equilibrium, the isothermal convention best describes transport; however, the non-equilibrium nature of the cellular interior motivates the consideration of non-isothermal conventions; the physically appropriate convention to use depends upon microscopic parameters and timescale hierarchies not captured in a coarse-grained model of diffusion. Note that while the Itô interpretation is deployed here, it is possible to convert from one interpretation to another (*Volpe and Wehr, 2016*) resulting in different interpretations converging at the same physical outcome (see *Appendix*). The equation used here is distinguished from the conventional 1D diffusion equation which arises from Fick's laws and is only unambiguously true for homogeneous diffusion (characterized by constant diffusivity).

Over the course of the simulation, particles accumulated in the low-diffusivity zone (*Figure 1A and C*), consistent with steady-state closed form Itô-convention solutions (*Tupper and Yang, 2012*). This accumulation entailed the transient depletion of particles on the high-diffusive side of the interface.

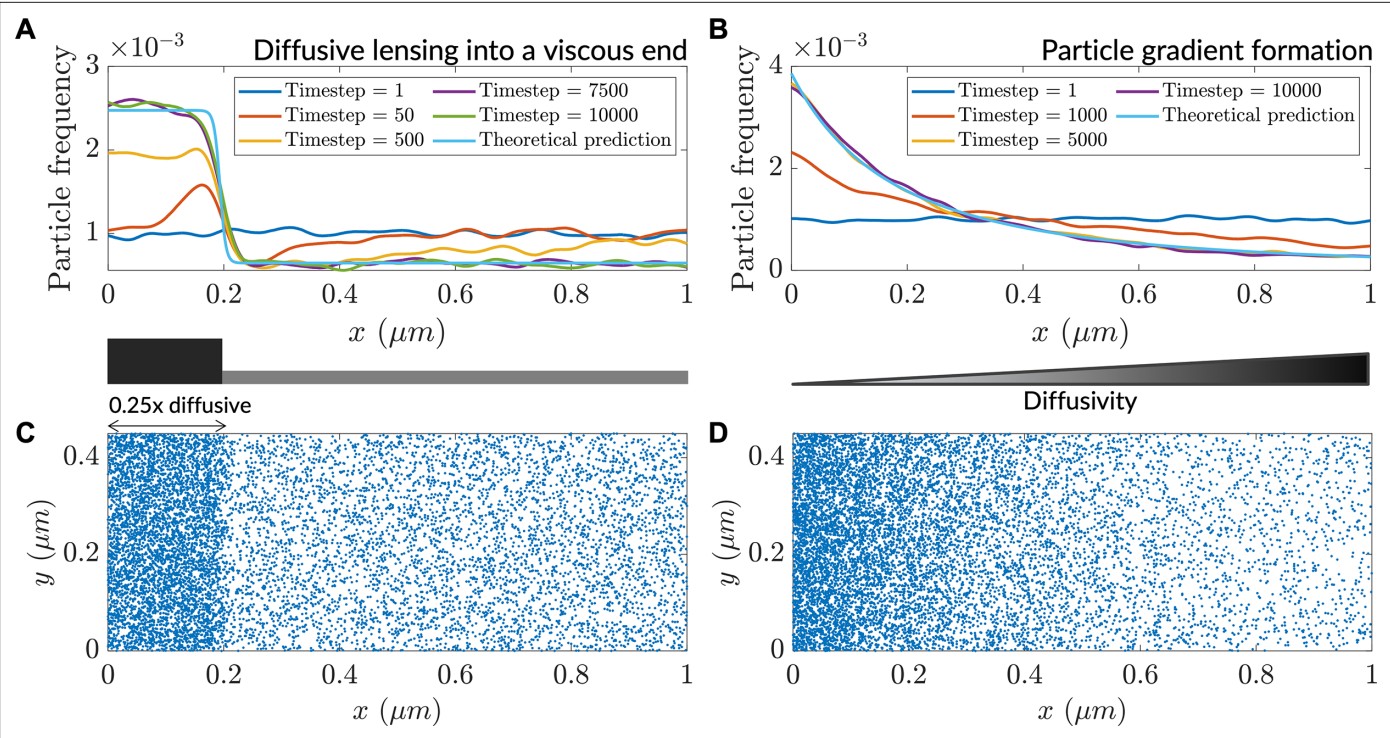

**Figure 1.** Low diffusivity leads to accumulation of particles. (**A**) Particle distribution at various timesteps of a simulation with a step-like lower-diffusivity region. (**B**) Particle distribution at various timesteps for a simulation with a diffusivity gradient. (**C**) Steady-state particle distribution for the simulation in (**A**). (**D**) Steady-state particle distribution for the simulation in (**B**).

The online version of this article includes the following figure supplement(s) for figure 1:

**Figure supplement 1.** Itô convention leads to Fokker-Planck diffusion, contrasting canonical ('Fickian') homogenization.

A similar accumulation was observed in a smooth diffusivity gradient (*Figure 1B and D*). In both cases, the results from agent-based modeling were corroborated by predictions of the steady-state analytical forms derived from theory. Thus, agent-based simulations demonstrate that under the Itô convention, areas of decreased diffusivity lead to increases in the concentration of diffusing particles. We term this phenomenon 'diffusive lensing'.

## Interaction-mediated clustering is affected by heterogenous diffusivity

Diffusive lensing is an interaction-free mode of concentrating particles that stands in contrast to a more typical paradigm of particle accumulation: interaction-driven formation of higher-order structures like protein complexes, gels, crystals, and phase-separated condensates (*Banani et al., 2017*; *Vekilov, 2010*; *Wu et al., 2023*). How might interaction-induced clustering be modulated by inhomogeneous diffusion in a cellular context? To address this question, we heuristically modeled interparticle interactions via a neighbor-sensing scheme in high and low interaction-strength regimes. The scheme involved using a step size for the modeled particle, which decreases as the number of particles in the vicinity increases (see *Methods*). At low interaction strength, clustering occurred only at the low-diffusivity end of a gradient (*Figure 2A*), while the same interaction strength was insufficient to produce clusters in a uniform diffusivity distribution (*Figure 2—figure supplement 1A and C*). In contrast, a high interaction strength resulted in robust clustering manifesting before particle gradient formation reached the steady-state, leading to clustering towards the high-diffusivity side of the simulation region as well (*Figure 2B*). At this high interaction strength, the clustering rate remained the same throughout the region in the absence of a gradient (*Figure 2—figure supplement 1B and D*). Taken together, the results reveal that diffusive lensing can modulate clustering and under certain circumstances cause diffusivity-dependent localized cluster formation, and furthermore that the relative strengths and timescales of each phenomenon quantitatively dictate whether increased clustering will preferentially occur in low-diffusive zones. Similar density-dependent clustering is observed in the

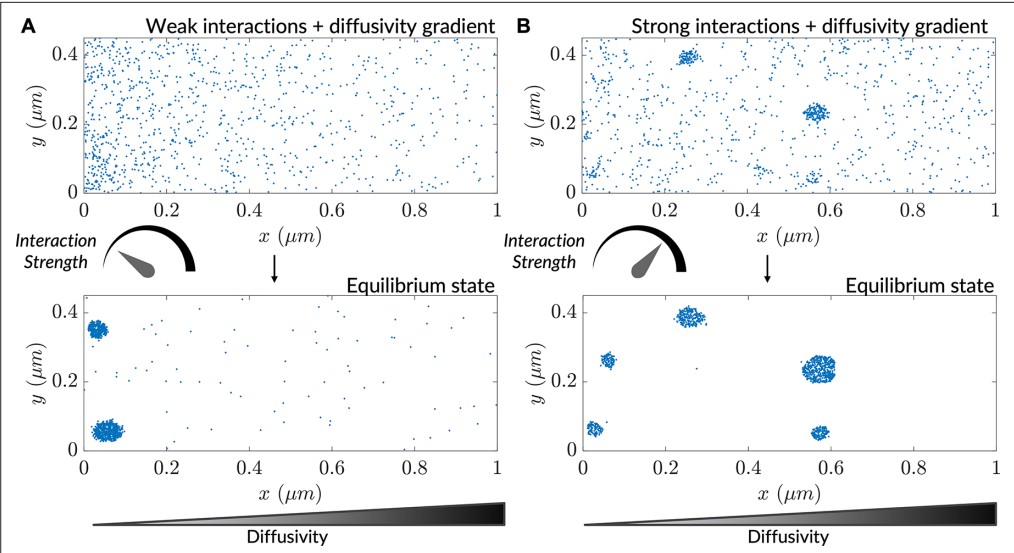

**Figure 2.** Interaction-driven clustering is modulated by heterogenous diffusivity. (**A**) Progress of a simulation comprising particles possessing weak interactions ($k = 0.04$ is the interaction strength; see *Methods*), initialized with a uniform concentration of particles. (**B**) Progress of a simulation comprising particles possessing strong interactions ($k = 0.1$), initialized with a uniform concentration of particles.

The online version of this article includes the following figure supplement(s) for figure 2:

**Figure supplement 1.** Particle clustering at different strengths in homogeneous versus heterogeneous diffusivity environments.

case of active Brownian particles during motility-induced phase separation (*Cates and Tailleur, 2015*). Effects of diffusive lensing on particle concentration may additionally regulate reaction rates and drive stochastic clustering of enzymes (*Jilkine et al., 2011*).

## Heterogeneous diffusion alters bulk particle motion as measured by *in silico* microrheology

The diffusion coefficient is a fundamental biophysical parameter that affects numerous other phenomena, including biochemical reaction rates. To elucidate particle diffusion at the microscale in the context of diffusive lensing, we used an in silico implementation of microrheology to analyze particle trajectories (see *Methods*; *Figure 3—figure supplement 1A*). We computed the mean squared displacements (MSDs) for uniform diffusivity simulations (in the case of unencumbered and confined diffusion) and used these to understand how MSD is affected by heterogenous diffusivity in two cases: a continuous diffusivity gradient and a discrete step in diffusivity.

Particle diffusion was unencumbered in the case of large bounds (relative to step size) (*Figure 3A*) and confined in the case of small bounds (*Figure 3B*) all in agreement with earlier results (*Dix and Verkman, 2008*; *Saxton, 2007*). The MSD at saturation in homogeneously diffusive systems was found to be agnostic to the underlying uniform diffusivity of the system, indicating that it is exclusively determined by the simulation region size. In contrast, particles in a diffusivity gradient exhibited dynamics intermediate to those of homogeneous high and low diffusivity cases, both in the diffusion coefficient and saturation MSD (*Figure 3C*, inset). The lowering of the saturation MSD reflects particle diffusion occurring within apparent simulation region bounds that confine more than the actual simulation region size. We note that such effective modifications of geometry are also a general feature of optical lensing. Apparent bounds were also found to occur in the two-zone diffusivity case (as in *Figure 1A*) where, at steady-state, particles populated the simulation region non-uniformly (*Figure 3—figure supplement 1B*). For most of the diffusivity ratio parameter space, irrespective of whether the smaller zones were more or less diffusive relative to the bulk, a reduction in MSD was seen indicating effectively lower diffusion bounds (*Figure 3D*). The magnitude of reduction depended on whether most particles resided in the larger or smaller of the two zones. In one observed case ($\frac{\mu_i}{\mu_o} = 0.25$), however, the saturation MSD was higher than what was seen in the homogeneous diffusion scenario possibly

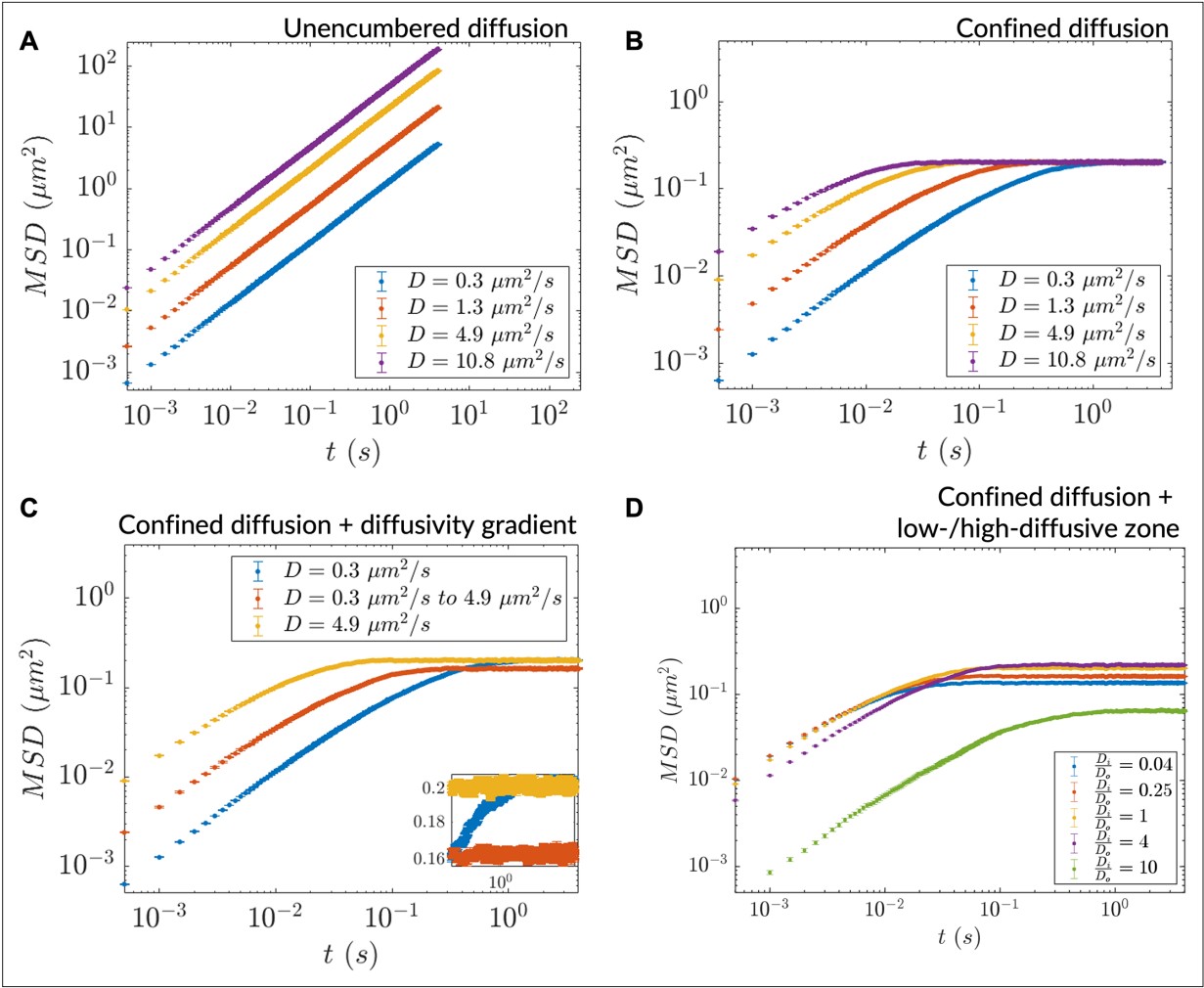

**Figure 3.** Heterogeneous diffusion alters bulk particle motion as measured by in silico microrheology. (**A**) mean squared displacement (MSD) versus time for homogeneous diffusion of 10,000 particles in a 5 mm × 5 mm simulation region. (**B**) Same as (**A**) for homogeneous diffusion in a more tightly bounded simulation region (1 μm × 0.45 μm). (**C**) MSD versus time for inhomogeneous diffusion in a diffusivity gradient versus homogeneous diffusion in the extreme diffusivity cases (simulation region size: 1 μm × 0.45 μm). Inset: zoomed region showing differential saturation of the MSD. (**D**) MSD versus time for inhomogeneous diffusion due to a stepwise diffusivity distribution with diffusivity ratio $\frac{\mu_i}{\mu_o}$ relative to the bulk (simulation region size: 1 μm × 0.45 μm). In all cases, n=10,000 particles for MSD calculation (error bars denote SEM).

The online version of this article includes the following figure supplement(s) for figure 3:

**Figure supplement 1.** Magnitude and distribution of inhomogeneity in diffusivity affects diffusive lensing.

due to particles robustly populating the bulk milieu followed by directed motion into the low-diffusive zone. The saturation MSD was also found to depend on the location of the low-diffusive zone: a more-centered zone resulted in a lowered saturation value, possibly due to weaker ratchet effects (*Figure 3—figure supplement 1C and D*). These results point to the insufficiency of using the diffusion coefficient alone to describe diffusion in heterogenous milieu. They also indicate a potentially rich interplay between heterogenous diffusivity and anomalous diffusion that requires further investigation.

## In silico FRAP in heterogeneously diffusive environments reveals drivers of mesoscale dynamics

The in silico microrheology analysis we performed provided insights into dynamics at the single-particle level (i.e. the microscale). To explore collective, emergent behaviors at the mesoscale while continuing to make contact with feasible experiments, we employed an in silico version of fluorescence recovery after photobleaching (in silico FRAP) (*Figure 4—figure supplement 1A and B*), in

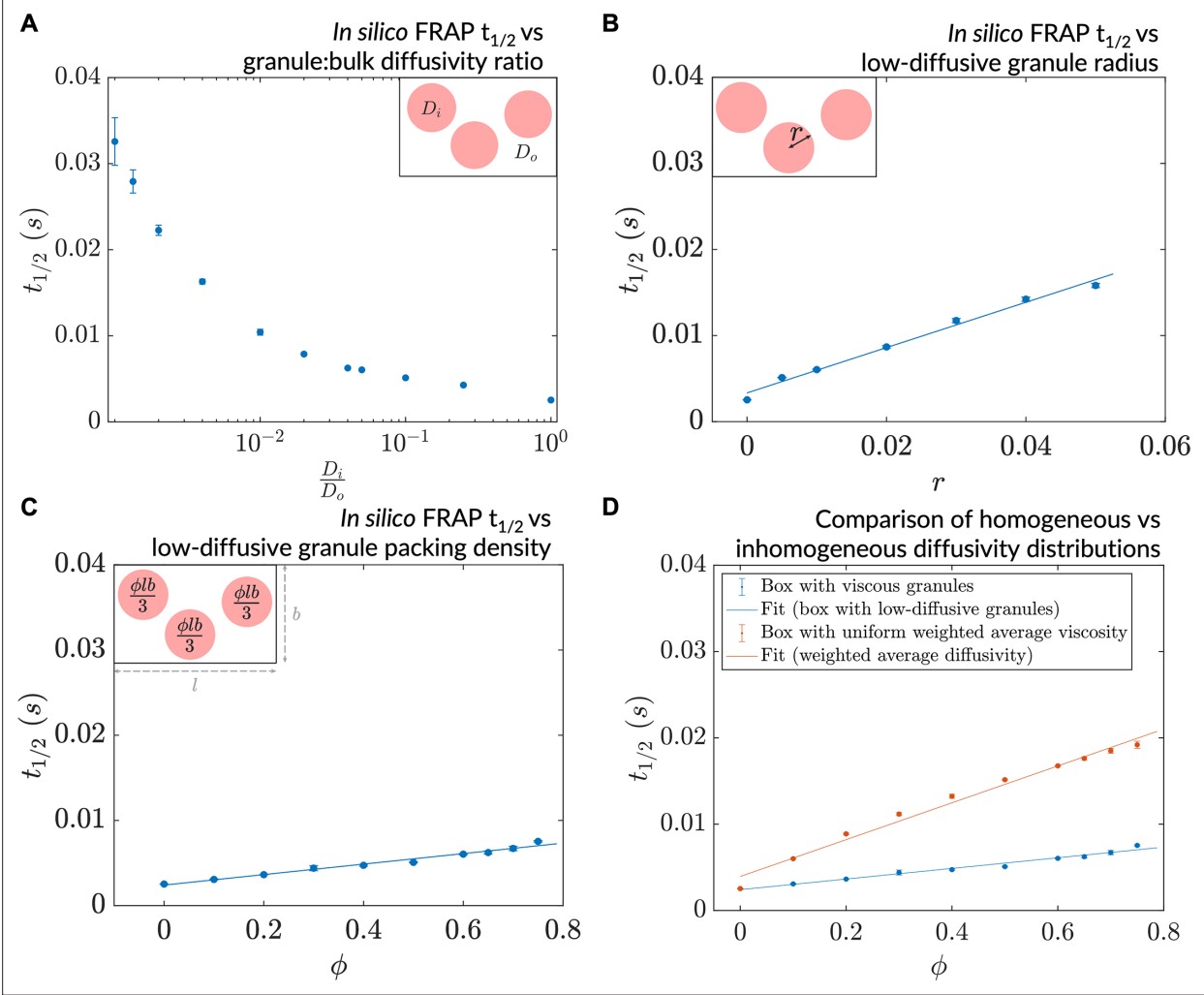

**Figure 4.** A decrease in granule diffusivity, an increase in granule radius, or packing density slows down mesoscale dynamics. (**A**) Simulated fluorescence recovery after photobleaching (FRAP) $t_{1/2}$ as a function of granule:bulk diffusivity ratio ($r = 0.1\ \mu m$, $\phi = o.6$). (**B**) Simulated FRAP as a function of granule radius ($\frac{\mu_i}{\mu_o} = 0.05$, $\phi = 0.6$). (**C**) Simulated FRAP $t_{1/2}$ as a function of granule packing density ($\frac{\mu_i}{\mu_o} = 0.05$, $\phi = 0.01\ \mu m$). (**D**) Simulated FRAP for homogeneous and inhomogeneous diffusivity setups realizing the same effective diffusivities ($\frac{\mu_i}{\mu_o} = 0.05$, $\phi = 0.01\ \mu m$). In all cases, n=3 ROIs were chosen for the simulated photobleaching (error bars denote SEM).

The online version of this article includes the following figure supplement(s) for figure 4:

**Figure supplement 1.** Dwell times for particles in low-diffusive granules dictate fluorescence recovery after photobleaching (FRAP) kinetics.

more cell-like inhomogeneous environments. In particular, we modeled low-diffusive patches/granules in a cell using a three-parameter disc-packing setup comprising granule radius ($r$), packing density ($\phi$), and the ratio of granule diffusivity to bulk diffusivity ($\frac{\mu_i}{\mu_o}$) (see *Methods*). We investigated the effect on dynamics of varying these parameters individually, with the goal of gaining an understanding of the effects of varying the amount, nature, and distribution of viscogens in cells. In all cases, the in silico 'photobleaching' event was conducted after the steady-state was attained (**Figure 4—figure supplement 1C, D and E**). To explain observed changes in the recovery time that would be measured in a FRAP-type experiment, we probed how the mean dwell time of particles in low-diffusive granules varies as a function of these parameters. A decrease in the diffusivity ratio ($\frac{\mu_i}{\mu_o}$) at fixed $\phi$ and $r$ resulted in a decline in measured particle mobility, as characterized by an increase in the simulated FRAP $t_{1/2}$ values (**Figure 4A**). Decreasing $\frac{\mu_i}{\mu_o}$ from 1 to 0.1 caused an approximate doubling of $t_{1/2}$ (or halving of diffusivity). Similar reduction in mobility was observed upon variation of $\phi$ or $r$ separately, keeping the diffusivity ratio constant (**Figure 4B and C**). The decrease in average mobility in all three cases arose from changes in flux between the low-diffusive and bulk zones, as reflected by an increase in mean

dwell times of particles within low-diffusive granules (*Figure 4—figure supplement 1F, G and H*). Furthermore, such reductions in mobility were emergent in that they arose from the interplay between granular diffusivity and bulk-granule fluxes, as the regions of interest in the simulated photobleaching events comprised granules and the surrounding bulk environment. To investigate whether particle dynamics is affected by the underlying topography realizing the system's diffusivity, we averaged the granular and bulk diffusivity values to produce weighted-average diffusivity values, and compared in silico recovery in these simulations to that of the equivalent granule-comprising simulations. Such an averaging of the diffusivity to cause an effective uniform mobility for all resident particles resulted in slower dynamics than that of the equivalent granule-comprising simulations (*Figure 4D*). We conclude that inhomogeneity in diffusivity drives rapid effective dynamics via fluxes between the granular (interior) and bulk (exterior) environments, creating 'diffusive highways' for particles to move rapidly between low-diffusive regions. The diffusive lensing of particles into low-diffusive zones, and their consequent dwelling in these regions, can be tuned by modulating the underlying diffusivity distribution in myriad ways.

## Discussion

The complex milieu of the cellular interior has been recently shown to feature heterogeneous diffusivity (*Garner et al., 2023*; *Huang et al., 2022*; *McLaughlin et al., 2020*; *Parry et al., 2014*; *Śmigiel et al., 2022*; *Xiang et al., 2020*), yet the consequences of such inhomogeneity on compartmentalization and mesoscale molecular dynamics have remained unclear. Through agent-based modeling of diffusion using the Itô integration convention, we show that heterogenous diffusivity can lead to simulated particle trajectories converging into low-diffusive hotspots, causing the accumulation of diffusing particles into membrane-less compartments defined by the lower-diffusivity zones. We term this mode of transport 'diffusive lensing.' The underlying conclusions from our 2D simulations extend to 3D directly (see *Methods*). Diffusive lensing has wide-ranging effects on particle distribution and dynamics and, furthermore, it can occur across a wide parameter space. We, therefore, speculate that diffusive lensing is a ubiquitous phenomenon in living systems.

We found that inhomogeneous diffusivity allows for particle mobility at the microscale and mesoscale to be different from that expected in the presence of homogeneous diffusion. Such an expectation is in line with predicted and observed deviations from normal diffusion in cells (*Bancaud et al., 2012*; *Baum et al., 2014*). The relative strengths of diffusive lensing and inter-particle interactions (if any) determined the extent to which clustering was modulated by diffusive lensing: this interplay may be important for determining the effects of inhomogeneous diffusivity on biochemical reaction rates. In these simulations of clustering, particle concentration did not affect diffusivity. In the case that particle concentration decreases diffusivity (for example in the case of branched polysaccharides like glycogen), diffusive lensing may create a positive feedback loop that drives particles into areas where low diffusivity has been nucleated. The effect of diffusive lensing on runaway pathological processes like protein aggregation is a potential direction for future work.

Spatially-averaged effective diffusion timescales were found to depend on the microscopic diffusivity distribution: the same average diffusivity can give rise to slower or faster dynamics depending on whether it is realized via homogeneous or heterogenous diffusivity distributions. In the latter case, the bulk region interspersed between the low-diffusive hotspots provides 'diffusive highways' that contribute to large fluxes at the diffusivity interface, thereby accounting for the faster dynamics. Such expressways and their associated fluxes may impact reaction kinetics by altering substrate turnover rates, congruent with the model of unusual transport processes potentially modifying reaction kinetics (*Bénichou et al., 2010*). In the context of subcellular low-diffusive regions (*Garner et al., 2023*), cells may compensate for geometry-imposed constraints on packing density and size of these regions by altering the diffusivity ratio (against the bulk milieu) instead. To map the detailed effects of inhomogeneous diffusivity on reaction rates, however, our work suggests that a key prerequisite is to chart a suitable set of meta parameters that provide an adequate description of inhomogeneous diffusion (*Jin and Verkman, 2007*), as a one-parameter description relying exclusively on the average diffusion coefficient is insufficient to fully specify the dynamics.

Changes in viscosity have been shown to occur in the context of cellular processes including cell death (*Kuimova et al., 2008*), stress adaptation (*Persson et al., 2020*) and protein aggregation (*Thompson et al., 2015*). At any given time point, intracellular transport dynamics arise emergently

from contributions across length scales ranging from crowding in the bulk milieu due to proteins (*Wang et al., 2010*), and large biomolecules (*Delarue et al., 2018*) to cytoskeleton (*Carlini et al., 2020*; *Chaubet et al., 2020*) and active flows in the cytoplasm (*Arcizet et al., 2008*), all leading to unusual anomalous diffusive behaviors at the mesoscale (*Banks and Fradin, 2005*; *Bressloff, 2014*; *Dix and Verkman, 2008*; *Höfling and Franosch, 2013*; *Kuznetsova et al., 2015*; *Swaminathan et al., 1997*; *Zhou et al., 2008*). These diffusive behaviors cannot be decoupled from the intrinsic heterogeneity in biomolecular properties themselves (*Heald and Cohen-Fix, 2014*; *Milo and Phillips, 2015*). The effects of all of these subcellular determinants and energy-dependent processes on how position-dependent diffusivity is maintained in a cell remains unclear.

Not all cases of heterogeneous diffusivity will lead to diffusive lensing. This ambiguity is captured by the so-called Itô-Stratonovich dilemma (*Lau and Lubensky, 2007*; *Sokolov, 2010*; *Tupper and Yang, 2012*; *Van Kampen, 1988*; *Volpe and Wehr, 2016*). Any mathematical conceptualization of diffusion in the presence of position-dependent diffusivity must confront this dilemma, according to which the steady-state concentration distribution of a diffusing tracer depends not only on the localized diffusivity distribution but also on conventions based on microscopic parameters not captured in a coarse-grained model of diffusion; these parameters might, for example, include correlation lengths and times of viscogens or physical characteristics of polymers (*Bo et al., 2021*; *Kupferman et al., 2004*; *Lau and Lubensky, 2007*; *Sokolov, 2010*; *Tupper and Yang, 2012*; *Van Kampen, 1988*; *Vishen et al., 2019*). We speculate that any source of heterogeneity in diffusivity (including, but not limited to: mesh size experienced by the diffusing tracer, temperature changes, viscogen identity, and concentration) can, in turn, modulate diffusive lensing by means of altering either the particle or the environment-induced noise relaxation time. While the Itô convention is deployed here to model the nonequilibrium cellular interior (*Gnesotto et al., 2018*; *Phillips et al., 2012*), in some cases the isothermal convention may be better suited for modeling transport. The choice of the convention (and the effect of the dilemma, by extension) may also be subverted by recasting the dynamics into an alternate convention by taking suitable drift terms into consideration (see *Appendix*). Indeed, diverse conventions have been used to model experimentally observed accumulation arising from varied sources of such position-dependent noise (*Bringuier, 2011*; *Pesce et al., 2013*; *Volpe and Wehr, 2016*).

Our work underscores the need to not only examine diffusivity distributions in vivo as a function of local composition and the environment, but also to study their time evolution in response to external stimuli. More speculatively, we suggest that diffusive lensing serves as a potential candidate for a rudimentary mode of pre-biotic compartmentalization. Lensing-driven accumulation of diverse biomolecules may have served to produce chemically enriched spaces, acting as an antecedent of more sophisticated, membrane-bound, and membraneless organizational modalities; such a proto-cell organization is orthogonal to currently studied models (*Monnard and Walde, 2015*). This work demonstrates that diffusive lensing can have strong effects on transport and may be common in cellular contexts, modulating both passive and active flows. Future experimental and theoretical work will elucidate the extent of lensing inside and outside of cells and its effects on the biochemical reactions that sustain life.

## Methods
### Agent-based modeling (random walk simulations)

Agent-based modeling of diffusion was conducted via 2D random walk simulations. Irrespective of how the Itô-Stratonovich dilemma is interrogated, the underlying diffusion equations contain additive, separable contributions from each dimension, and this extends to 3D as well. Calculations were, therefore, carried out in 2D for simplicity and visualizability. Non-interacting point particles were initialized uniformly in a 2D simulation region with an aspect ratio matching that of an *E. coli* bacterium (*Phillips et al., 2024*). During each time step (also termed epoch or frame), every particle was moved along each coordinate by step sizes sampled from a uniform distribution, $U(-S, S)$, where denotes the step size limit. Across a large number of steps, the distribution of displacements converges to the normal distribution by virtue of the central limit theorem. While sampling was not performed via the normal distribution directly by using the diffusion coefficient ($D$) as a parameter, the diffusion coefficient was instead arrived at as an emergent property of trajectories comprising a simulation, in a

ground-up fashion. Reflection off the wall was modeled using a mirror-image rule. To model a zone of differential diffusivity relative to bulk diffusivity (either a fluid or a diffusivity zone), particle step sizes were sampled from zones characterized by different diffusivities, noting that the diffusion coefficient and diffusivity are inversely related *Phillips et al., 2024* and $S \infty \sqrt{D}$. At all times, step sizes were sampled from distributions defined by the diffusivity around the present position in accordance with the Itô interpretation of multiplicative noise (*Volpe and Wehr, 2016*) (for theoretical predictions of the steady-state behaviors, see *Numerical methods for the diffusion equations*). In all simulations, a set seed of 1 was used for the random number generator. Simulations were run on MATLAB R2020a on Sherlock (a high-performance computing cluster at Stanford).

In the simulations which included inter-particle interactions, these interactions were modeled via a neighbor-sensing approach. The step size limit was modified as per the relation, $S_{eff} = S e^{-kn}$, where-denotes the sensing strength and denotes the number of neighbors (defined as those particles lying within a cutoff span around the particle in question). Such a rule-based approach modeled an effective attractive potential for the inter-particle interactions. Local density calculation used the same cutoff and the data were normalized to the mean local density of particles during initialization. Considering the computational work due to neighbor-sensing, a smaller number of particles ($10^3$) were deployed, for a longer period of $2 \times 10^4$ epochs.

In the low-diffusive granule simulations, the granules were modeled as disks with randomly initialized centers and fixed radii ($r$), covering the simulation region up to a desired packing density, $\phi$. The algorithm saturated for $\phi \geq 0.6$, in which case, the disks were generated as per cubic close packing, and their positions were incrementally deviated over steps to reduce local ordering as much as possible. The ratio of diffusivity inside the granules to diffusivity outside the granules ($\frac{\mu_i}{\mu_0}$) was the third parameter under consideration. No two disks were allowed to overlap and all disks were kept confined within the boundaries of the simulation region. The default setup is as follows: $r = 0.01$ μm (uniform), $\phi = 0.6$ (that is, 60% of the simulation region is covered by the granules) and $\frac{\mu_i}{\mu_0} = 0.05$. Titration of one of these three parameters involved keeping the other two at the specified levels.

## Numerical methods for the diffusion equations

The Fokker-Planck equations corresponding to the Ito, Stratonovich, and isothermal interpretations of inhomogeneous diffusion are as follows *Gardiner, 2004*; *Tupper and Yang, 2012* (here $c(x, t)$ denotes the concentration distribution and $D(x)$ denotes the position-dependent diffusivity):

Itô interpretation: $\frac{\partial c}{\partial x} = \frac{\partial^2 cD}{\partial x^2}$
Stratonovich interpretation: $\frac{\partial c}{\partial x} = \frac{\partial}{\partial x}(\sqrt{D} \frac{\partial c \sqrt{D}}{\partial x})$
Isothermal interpretation: $\frac{\partial}{\partial x} = \frac{\partial}{\partial x}(D \frac{\partial c}{\partial x})$

These equations were numerically evaluated via forward time-centered space (FTCS) schemes, with length and time increments set as $10^{-3}$ and $5 \times 10^{-7}$ arbitrary units, respectively, and the number of time steps was set to $10^5$. A Gaussian well profile was used for the diffusion coefficient and the initial condition for the concentration distribution was a uniform distribution (*Figure 1—figure supplement 1B*). For the theoretical prediction in each case, the following relation is used: $c(x, t)D(x)^{1-\alpha} = constant$ in steady-state, where $\alpha$ denotes the integration convention used (*Tupper and Yang, 2012*). Analysis and data visualization were performed on MATLAB R2019a.

## In silico microrheology

Analysis of particle trajectories was carried out via quantifying the mean squared displacements (MSD). These were calculated from $10^4$ trajectories (each $10^5$ timesteps in duration) per simulation. The timestep was set as 50 μs so that the diffusion coefficient was $\approx 5 \, \mu m^2/s$ order of magnitude for a small protein's mobility in the *E. coli* cytoplasm (*Milo and Phillips, 2015*).

## In silico FRAP

In silico fluorescence recovery after photobleaching (FRAP) studies were performed on the diffusion simulations to quantify emergent dynamics at the mesoscale. $10^5$ particles were deployed for a total duration of 0.5 s ($10^4$ epochs). Circular regions (radius of 0.2 μm) were chosen as the regions of interest (ROIs). In silico photobleaching was instantaneously performed and involved assigning the particles in the ROI the photobleach status. The background was chosen from a uniform diffusivity

setup to ensure that the normalization is standardized. The outward turnover of these particles and the simultaneous inward flux of unbleached particles were captured via $t_{1/2}$, the time taken for recovery up to 50% of the steady-state level of unbleached particles in the ROI (*Sprague and McNally, 2005*). In these simulations, $t_{1/2}$ connotes the time taken for the number of 'unbleached' particles in the ROI to reach 50% of the steady-state value. To dissect particles' behavior during the simulation (in terms of bias towards inhabiting the low-diffusive granules), we calculated the mean dwell time across all particles, per simulation. This involved averaging the periods (of any duration) spent by particles inside low-diffusive granules. For normalization, the total simulation duration was used (0.5 s).

## Acknowledgements

We thank the Brandman Lab, Grant M Rotskoff, J E Ferrell Jr., Z Dogic, A Chaudhuri, and G Chu for helpful discussions. We thank P Guptasarma for helpful discussions and for facilitating the arrangement between IISER Mohali, UCSB, and Stanford. Simulations conducted in this study were run on the Sherlock high-performance computing cluster maintained by the Stanford Research Computing Center. ARV. is supported by the KVPY fellowship. K.H.L is supported by the NSF Graduate Research Fellowship Program. O B is funded by the National Institutes of Health grant R35GM153301.

## Additional information

### Funding

| Funder | Grant reference number | Author |
|---|---|---|
| National Institutes of Health | R35GM153301 | Onn Brandman |
| National Science Foundation | DGE-1656518 | Kathy H Le |

The funders had no role in study design, data collection and interpretation, or the decision to submit the work for publication.

### Author contributions

Achuthan Raja Venkatesh, Conceptualization, Software, Formal analysis, Investigation, Visualization, Methodology, Writing - original draft, Writing - review and editing; Kathy H Le, Conceptualization, Investigation, Methodology, Writing - original draft, Writing - review and editing; David M Weld, Onn Brandman, Conceptualization, Resources, Formal analysis, Supervision, Investigation, Writing - original draft, Project administration, Writing - review and editing

### Author ORCIDs

Achuthan Raja Venkatesh (ID) http://orcid.org/0009-0003-1868-4068
Kathy H Le (ID) http://orcid.org/0009-0001-5575-7153
David M Weld (ID) http://orcid.org/0000-0002-4574-9491
Onn Brandman (ID) http://orcid.org/0000-0002-2084-154X

Reviewer #1 (Public review): https://doi.org/10.7554/eLife.89794.3.sa1
Reviewer #2 (Public review): https://doi.org/10.7554/eLife.89794.3.sa2
Author response https://doi.org/10.7554/eLife.89794.3.sa3

## Additional files

### Supplementary files
• MDAR checklist

## Data availability

The current manuscript is a computational study, so no data have been generated for this manuscript. Modelling code is uploaded as supplementary material and at https://zenodo.org/records/7957931.

The following dataset was generated:

| Author(s) | Year | Dataset title | Dataset URL | Database and Identifier |
|---|---|---|---|---|
| Achuthan RV, Kathy L, David W, Onn B | 2023 | Diffusive Lensing Code | https://doi.org/10.5281/zenodo.7957931 | Zenodo, 10.5281/zenodo.7957931 |

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

# Appendix

## Itô-Stratonovich dilemma and derivation of the generalized flux for heterogeneous diffusion

The Itô-Stratonovich dilemma is an ambiguity which arises directly from any analytical attempt to solve a stochastic differential equation comprising multiplicative (or position-dependent) noise (**Amir, 2020**; **Gardiner, 2004**; **Van Kampen, 1988**; **Volpe and Wehr, 2016**). The underlying mathematical reason for the ambiguity is the non-differentiable nature of the stochastic term, which causes different Riemann sum conventions to give quantitatively different results (**Gardiner, 2004**). From a more physical point of view, the different conventions correspond to different hierarchies of 'small' scales (correlation times and lengths of driving terms or viscogens, for example; see **Pesce et al., 2013** for a discussion of competing timescales, **Sokolov, 2010** for arriving at a context-dependent choice of interpretation). While the dilemma exists at the level of choosing an interpretation, it does not negate the appearance of diffusive lensing; indeed, only the magnitude of lensing is altered by the choice of the interpretation. This, too, may be subverted by adding an appropriate drift term to recast a stochastic differential equation (SDE) into one abiding by an alternate convention, as detailed later in this section. In summary, even the effect of the convention on the magnitude of diffusive lensing may be altered by recasting the SDE.

Formally, the dilemma can be boiled down to a choice of the parameter, $\alpha \left( 0 \leq \alpha \leq 1 \right)$, in a diffusion equation like

$$\frac{\partial c}{\partial t} = \frac{-\partial J}{\partial x} = \frac{\partial}{\partial x} \left[ \left( 1 - \alpha \right) c \frac{dD}{dx} + D \frac{\partial c}{\partial x} \right],$$

where $c(x, t)$ denotes the concentration distribution and $D(x)$ denotes the position-dependent diffusivity (see below for the derivation of the generalized flux). Clearly, for a position-dependent diffusivity, different values of $\alpha$ will give rise to different physical predictions. In thermal equilibrium, $\alpha$ must be equal to 1, but away from equilibrium it can take on different values depending on microscopic details of the physical system (**Gardiner, 2004**; **Kupferman et al., 2004**; **Pesce et al., 2013**; **Smythe et al., 1983**; **Turelli, 1977**; **Vishen et al., 2019**; **Wang et al., 2020**). Living cells are inherently removed from equilibrium by energy-driven processes breaking detailed balance (**Gnesotto et al., 2018**; **Phillips et al., 2012**). This motivates the calculations in this work, which use the Itô ($\alpha = 0$) integration convention, but in fact qualitatively similar results would be observed for any value of $\alpha$ other than 1. Taken together with the ability to recast an SDE into one confining to an alternate convention, the dilemma does not take away from our principal thesis on how low diffusivity zones can accrete particles.

Consider a diffusing tracer whose trajectory is specified by the stochastic process, $X\left(t\right)$. The stochastic process is continuous but is nowhere differentiable: the simplistic case of such a mathematical object is the Wiener process $W\left(t\right)$, also known as Brownian motion (**Gardiner, 2004**). Note that in general, however, the stochastic process is characterized by a deterministic drift $a\left(x, t\right)$ and diffusion $b\left(x, t\right)$. Indeed, $X\left(t\right)$ is in fact the solution of the stochastic differential equation (SDE):

$$dX\left(t\right) = a\left(x, t\right) dt + b\left(x, t\right) dW\left(t\right),$$

where $dW\left(t\right) = \eta\left(t\right) dt$ denotes the Wiener increment and $\eta\left(t\right)$ is the Gaussian white noise.

The stochastic differential equation here comprises multiplicative noise: that is, a position-dependent function, $b\left(x, t\right)$, factors into the noise term of the SDE. This necessitates confronting the Itô-Stratonovich dilemma; we consider a general convention $\alpha \in \left[0, 1\right]$, where $\alpha = 0, 0.5, 1$ correspond to the Itô, Stratonovich, and isothermal conventions, respectively.

The SDE detailed above corresponds to a Fokker-Planck equation describing the time-evolution of the probability distribution of the tracer, $p\left(x, t\right)$:

$$\frac{\partial p\left(x, t\right)}{\partial t} = \frac{-\partial}{\partial x} \left( a\left(x, t\right) p\left(x, t\right) + \alpha b\left(x, t\right) \frac{\partial b\left(x, t\right)}{\partial x} p\left(x, t\right) \right) + \frac{1}{2} \frac{\partial^2}{\partial x^2} \left[ b^2\left(x, t\right) p\left(x, t\right) \right]$$

(Substituting for $\alpha$ yields specific cases; see Eqn. 4.3.20 in **Gardiner, 2004**, Eqns. 2.2.25, 2.2.26 in **Bressloff, 2014**, Table 2 and Discussion in **Tupper and Yang, 2012**.) Setting $a\left(x, t\right)$ (drift) to zero, and $b^2\left(x, t\right)$ (diffusion) to $2D$, a constant, yields Brownian motion, i.e., the canonical 1D diffusion

equation which may also be derived from Fick's laws. Keeping drift as zero, but setting diffusion to $2D(x)$ (space-dependent) yields:

$$\frac{\partial p(x,t)}{\partial t} = \frac{-\partial}{\partial x}\left(\alpha\frac{dD(x)}{dx}p(x,t)\right) + \frac{\partial^2}{\partial x^2}\left[D(x)p(x,t)\right]$$

which describes our scenarios in question. This equation can be simplified as follows:

$$\frac{\partial p(x,t)}{\partial t} = \frac{-\partial}{\partial x}\left[\alpha\frac{dD(x)}{dx}p(x,t) - \frac{\partial}{\partial x}\left(D(x)p(x,t)\right)\right]$$

$$\Rightarrow \frac{\partial p(x,t)}{\partial t} = \frac{-\partial}{\partial x}\left[-\left(1-\alpha\right)p(x,t)\frac{dD(x)}{dx} - D(x)\frac{\partial p(x,t)}{\partial x}\right] = \frac{-\partial}{\partial x}J(x,t),$$

where $J = -\left(1-\alpha\right)p(x,t)\frac{dD(x)}{dx} - D(x)\frac{\partial p(x,t)}{\partial x}$ is the generalized flux. Re-writing in terms of concentration yields the final expression, $J = -\left(1-\alpha\right)c\frac{dD}{dx} - D\frac{\partial c}{\partial x}$. Substituting for $\alpha$ and calculating $\frac{-\partial}{\partial x}J(x,t)$ yields the equations deployed in *Methods: Numerical methods for the diffusion equations*.

Note that it is also possible to convert between different interpretations of the multiplicative noise term by adding suitable drift terms as detailed in *Volpe and Wehr, 2016*; *Tupper and Yang, 2012*. In cases where thermal equilibrium is relevant, the Itô- and Stratonovich-conforming equations may be modified by adding drift terms to conform to that of the isothermal interpretation. Conversely, it is also possible to convert an equation based upon the isothermal convention to an Itô- or Stratonovich-conforming equation by modifying the drift term in the latter. Potential use cases as well as the consequences of following such conversions are detailed in these three example prescriptions below:

**Appendix 1—table 1.** Converting between stochastic integration conventions.

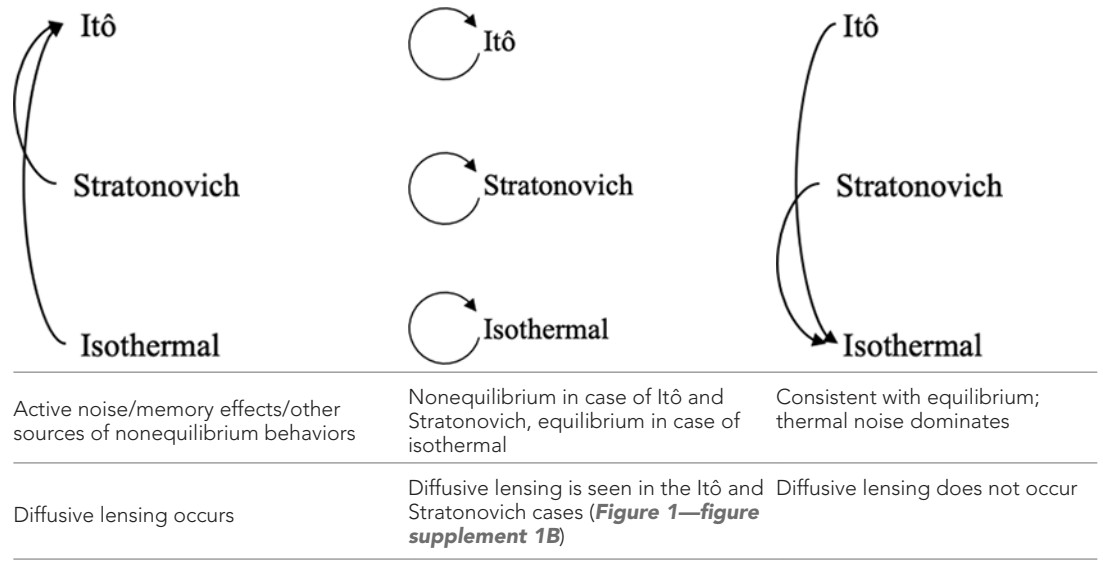

| | | |
|---|---|---|
| Active noise/memory effects/other sources of nonequilibrium behaviors | Nonequilibrium in case of Itô and Stratonovich, equilibrium in case of isothermal | Consistent with equilibrium; thermal noise dominates |
| Diffusive lensing occurs | Diffusive lensing is seen in the Itô and Stratonovich cases (*Figure 1—figure supplement 1B*) | Diffusive lensing does not occur |

