## [Editor Report · eLife assessment]

The authors discuss an effect, "diffusive lensing", by which particles would accumulate in high-viscosity regions – for instance in the intracellular medium. To obtain these results, the authors rely on agent-based simulations using custom rules performed with the Ito stochastic calculus convention. The "lensing effect" discussed is a direct consequence of the choice of the Ito convention without spurious drift which has been discussed before and its adequacy for the intracellular medium is insufficiently discussed and relatively doubtful. Consequently, the relevance of the presented results for biology remain unclear and based on **incomplete** evidence.

---

## [Referee Report · Reviewer #1 (Public review)]

The revised manuscript "Diffusive lensing as a mechanism of intracellular transport and compartmentalization" is very similar to the original manuscript. The main difference between the revised and the original manuscript is that the authors have removed the reference to viscosity gradient and instead talk of diffusivity gradient. With this change the manuscript the analysis and claims in the manuscript are much more aligned. The manuscript, as the original version, explores the role of spatially varying diffusion constant in three scenarios:

(i) Spatial localization of non-particles

(ii) Clustering in presence of inter-particle interactions

(iii) Moment analysis for non-interacting particles in space with discrete patches of inhomogeneous diffusivity.

Since the manuscript has not changed much the strengths and weaknesses, in my opinion, remain similar to that of the original manuscript.

Strengths: The implications of a heterogeneous environment on phase separation and reaction kinetics in cells are under-explored. This makes the general theme of this manuscript relevant and interesting.

Weaknesses: The central part of the paper "diffusive lensing", i.e., particles localizing in the region of low diffusion constant is not new. Some of the papers authors cite already show that. The parts on phase separation and frap analysis that could provide new results are not rigorous enough for a theory paper.

I reiterate some of my comments from the original version that are valid for the revised version as well.

My main criticism was not to say that some convention should be used or some not. But instead, the main point was to say that just because there is spatial diffusion constant that does not mean there will be a spatial gradient of particles. From the authors response to my comments, it is clear that they understand the subtilties around it and are aware of the relevant papers. However, a reader not familiar with this discussion may work under the impression that if there if there is a spatialy varying diffusion constant in cell there will be an accumulation of particles in the region of low diffusivity but that may not always be the case. Moreover, localisation of particles in the region of low diffusivity has been reported in many different context. Some of the papers that the author cite already show that. For example, in Rupprecht et al. 2018 non-isothermal interpretation is applied to the dynamics of objects inside cells.

Given that the central result is not new. The paper could still be of general interest to the biophysics community if the follow up sections (ii) Clustering in presence of inter-particle interactions and (iii) Moment analysis for non-interacting particles in space with discrete patches of inhomogeneous diffusivity were analysed rigorously.

---

## [Referee Report · Reviewer #2 (Public review)]

Summary:

The authors study through theory and simulations the diffusion of microscopic particles, and aim to account for the effects of inhomogeneous viscosity and diffusion - in particular regarding the intracellular environment. They propose a mechanism, termed "Diffusive lensing", by which particles are attracted towards low-diffusivity regions where they remain trapped. To obtain these results, the authors rely on agent-based simulations using custom rules performed within the Ito stochastic calculus convention, without drift. They acknowledge the fact that this convention does not describe equilibrium systems, and that their results would not hold at equilibrium - and discard these facts by invoking the facts that cells are out-of-equilibrium. Finally, they show some applications of their findings, in particular enhanced clustering in the low-diffusivity regions. The authors conclude that as inhomogeneous diffusion is ubiquitous in life, so must their mechanism be, and hence it must be important.

Strengths:

The article is well-written, clearly intelligible, its hypotheses are stated relatively clearly and the models and mathematical derivations are compatible with these hypotheses. In the appendices, the authors connect their findings to known results for classic stochastic differential equation formalisms.

Weaknesses:

This study is, in my opinion, deeply flawed. The main problem lies in the hypotheses, in particular the choice of considering drift-less dynamics in the Ito convention. It is regrettable that the authors choose to use agent-based custom simulations with little physical motivation, rather than a well-established stochastic differential equations framework.

Indeed, stochastic conventions are a notoriously tricky business, but they are both mathematically and physically well-understood and do not result in any "dilemma" [some citations in the article, such as (Lau and Lubensky) and (Volpe and Wehr), make an unambiguous resolution of these]. In the continuous-time limit, conventions are not an intrinsic, fixed property of a system, but a choice of writing; however, whenever going from one to another, one must include a corresponding "spurious drift" that compensates the effect of this change - a mathematical subtlety that is omitted in the article (except in a quick note in the appendix): in the presence of diffusive gradients, if the drift is zero in one convention, it will thus be non-zero in another. It is well established that for equilibrium systems obeying fluctuation-dissipation, the spurious drift vanishes in the anti-Ito stochastic convention; more precisely one can write in the anti-Ito convention

dx/dt = - D(x)/kT grad U(x) + sqrt(2D(x)) dW

with D(x) the diffusion, kT the thermal energy (which is space-independent at equilibrium), and dW a d-dimensional Wiener process. Equivalently one can write in the Ito convention:

dx/dt = - D(x)/kT grad U(x) + sqrt(2D(x)) dW + div D(x) (*)

where the latter term is the spurious drift arising from convention change. This ensures that the diffusion gradients do not induce currents and probability gradients, and thus that the steady-state PDF is the Gibbs measure (this form has been confirmed experimentally, for instance, for colloidal particles near walls, that have strong diffusivity gradients despite not having significant forces). It generalizes to near-equilibrium systems with non-conservative forces and/or temperature gradient in the form:

dx/dt = F(x) + sqrt(2D(x)) dW + div D(x) (**)

where the drift field F(x) encodes these forces. In some cases, it has been shown through careful microscopic analysis that one can have effectively a different form for the last term, namely

dx/dt = F(x) + sqrt(2D(x)) dW + alpha div D(x)

where alpha is a "convention parameter" that would be = 1 at equilibrium. For instance, in the Volpe and Wehr review this can occur through memory effects in robotic dynamics, or through strong fluctuation-dissipation breakdown. In a near-equilibrium system, this should be strongly justified, as the continuous-time dynamics with alpha \neq 1 and drift F would be indistinguishable from one with alpha = 1 and drift F + (1-alpha) div D: the authors would have the burden of proving that the observed (absence of) drift is indeed due to alpha\neq 1, rather than to much more common force fields F(x).

Here, without further motivation than the statement that cells are out-of-equilibrium, drifts are arbitrarily set to zero in the Ito convention, which is in (**) the equivalent to adding a force with drift $-div D$ exactly compensating the spurious drift. It is the effects of this arbitrary force that are studied in the article. The fact that it results in probability gradients is trivial once formulated this way (and in no way is this new - many of the references, for instance Volpe and Wehr, mention this). Enhanced clustering is also a trivial effect of this probability gradient (the local concentration is increased by this force field, so phase separation can occur). As a side note the "neighbor sensing" scheme to describe interactions is itself very peculiar and not physically motivated - it violates stochastic thermodynamics laws too, as detailed balance is apparently not respected. There again, the authors have chosen to disregard a century of stochastic thermodynamics in favor of a non-justified unphysical custom rule.

The authors make no further justification of their choice of driftless Ito simulations than the fact that cells are out-of-equilibrium, leaving the feeling that this is a detail. They make mentions of systems (eg glycogen, prebiotic environment) for which (near-)equilibrium physics should mostly prevail, and of fluctuation dissipation ("Diffusivity varies inversely with viscosity", in the introduction). Yet the "phenomenon" they discuss is entirely reliant on an undiscussed mechanism by which these assumptions would be completely violated (the citations they make for this - Gnesotto '18 and Phillips '12 - are simply discussions of the fact that cells are out-of-equilibrium, not on any consequences on the convention).

Finally, while inhomogeneous diffusion is ubiquitous, the strength of this effect in realistic conditions is not discussed. Even in the most "optimistic" case where alpha=0 would make sense (knowing that in the cellular context we are discussing thermal systems immersed in water and if energy consumption and metabolism were stopped alpha would relax back to 1), the equation (*) above shows that having zero ito drift is equivalent to having a potential countering the spurious drift, with value

U(x) = kT log(D(x) / D0 )

[I have assumed isotropic diffusion for simplicity here, so the div is replaced by a grad]. This means that the diffusion contrasts logarithmically compare to the chemical potential ones -- for instance a major diffusion difference of 100x is equivalent to 4.6kT in potential energy, a relatively modest effect. To prove that the authors' effect of "diffusive lensing" is involved in such a system, one would thus have to

1. observe strong spatial variations of the diffusion coefficient (this is doable, and was done before), AND

2. show that there is an enrichment of the diffusing species in the low-diffusion region inversely proportional to the diffusion, AND

3. show that this enrichment cannot be attributed to mild differences in potential energy, for instance by showing that if nonequilibrium energy consumption stops, the concentration fully homogenizes while the diffusion gradients remain.

If the authors were to successfully show all that in an experimental system, or design a theoretical framework where these effects convincingly emerge from physically realistic microscopic dynamical rules, they would have indeed discovered a new phenomenon. In contrast, the current article only demonstrates the well-known fact that when using arbitrary dynamical rules in heterogeneous diffusion simulations, one can get concentration gradients.

---

## [Author Response]

The following is the authors’ response to the original reviews.

**eLife assessment**
The authors discuss an effect, "diffusive lensing", by which particles would accumulate in high-viscosity regions, for instance in the intracellular medium. To obtain these results, the authors rely on agent-based simulations using custom rules performed with the Ito stochastic calculus convention. The "lensing effect" discussed is a direct consequence of the choice of the Ito convention without spurious drift which has been discussed before and is likely to be inadequate for the intracellular medium, causing the presented results to likely have little relevance for biology.

We thank the editors and the reviewers for their consideration of our manuscript. We argue in this rebuttal and revision that our results and conclusions are in fact likely to have relevance for biology. While we use the Itô convention for ease of modeling considering its non-anticipatory nature upon discretization (see [62] for the discretization schemes), we refer to Figure S1B to emphasize that diffusive lensing occurs not only under the Itô convention but across a wide parameter space. Indeed, it is absent only in the normative isothermal convention; note that even a stochastic differential equation conforming to the isothermal convention may be reformulated into the Itô convention by adding suitable drift terms, allowing for diffusive lensing to be seen even in case of the isothermal convention. We note in particular that the choice of the convention is a highly context-dependent one [52]; there is not a universally correct choice, and one can obtain stochastic differential equations consistent with Ito or Stratonovich interpretations in different regimes. Lastly, space-dependent diffusivity is now an experimentally well-recognized feature of the cellular interior, as noted in our references and as discussed further later in this response. This fact points towards the potential relevance of our model for subcellular diffusion.

In our revised preprint, we have made changes to the text and minor changes to figures to address reviewer concerns.

**Responses to the Reviewers**

We thank the reviewers for their feedback and address the issues they raised in this rebuttal and in the revised manuscript. The central point that the reviewers raise concerns the validity of the drift-less Itô interpretation in modeling potential nonequilibrium types of subcellular transport arising from space-dependent diffusivity. If the drift term were considered, the resulting stochastic differential equation stochastic differential equation (SDE) is equivalent to one arising from the isothermal interpretation of heterogeneous diffusivity (62), wherein no diffusive lensing is seen (as shown in Fig. S1B). That is, the isothermal interpretation and the drift-comprising Itô SDE produce the same uniform steady-state particle densities.

While we agree with the reviewers that for a given interpretation, equivalent stochastic differential equations (SDEs) arising from other interpretations may be drawn, we disagree with the generalization that all types of subcellular diffusion conform to the isothermal interpretation. That is, there is no reason why any and all instances of nonequilibrium subcellular particle diffusion must be modeled using isothermal-conforming SDEs (such as the drift-comprising Itô SDE, for instance). We refer to [52] which prescribes choosing a convention in a context-dependent manner. In this regard, we disagree with the second reviewer’s characterization of making such a choice merely a “choice of writing” considering that it is entirely dependent on the choice of microscopic parameters, as detailed in the discussion section of the manuscript. The following references have also been added to the manuscript: the reference from the first reviewer (32) proposes a prescription for choosing an appropriate convention based upon comparing the noise correlation time and the particle relaxation time. The reference notes that the Itô convention is appropriate when the particle relaxation time is large when compared to the noise correlation time and the Stratonovich convention is appropriate in the converse scenario. In (Rupprecht et al. 2018), active noise is considered and the resulting Fokker-Planck equation conforms to the Stratonovich convention when thermal noise was negligible. The related reference, (61) compares three timescales: those of particle relaxation, noise correlation and viscoelastic relaxation, to make the choice. Indeed, as noted in the manuscript, lensing is seen in all but one interpretation (without drift additions); only its magnitude is altered by the interpretation/choice of the drift term. The appendix has been modified to include a subsection on the interchangeability of the conventions.

Separately, with regards to the discussion on anomalous diffusion, the section on mean squared displacement calculation has been amended to avoid confusing our model with canonical anomalous diffusion which considers the anomalous exponent; how the anomalous exponent varies with space-dependent diffusivity offers an interesting future area of study.

Responses to specific reviewer comments appear below.

**Reviewer #1 (Public Review):**
The manuscript "Diffusive lensing as a mechanism of intracellular transport and compartmentalization", explores the implications of heterogeneous viscosity on the diffusive dynamics of particles. The authors analyze three different scenarios:(i) diffusion under a gradient of viscosity,(ii) clustering of interacting particles in a viscosity gradient, and(iii) diffusive dynamics of non-interacting particles with circular patches of heterogeneous viscous medium.The implications of a heterogeneous environment on phase separation and reaction kinetics in cells are under-explored. This makes the general theme of this manuscript very relevant and interesting. However, the analysis in the manuscript is not rigorous, and the claims in the abstract are not supported by the analysis in the main text.Following are my main comments on the work presented in this manuscript:(a) The central theme of this work is that spatially varying viscosity leads to position-dependent diffusion constant. This, for an overdamped Langevin dynamics with Gaussian white noise, leads to the well-known issue of the interpretation of the noise term.The authors use the Ito interpretation of the noise term because their system is non-equilibrium.One of the main criticisms I have is on this central point. The issue of interpretation arises only when there are ill-posed stochastic dynamics that do not have the relevant timescales required to analyze the noise term properly. Hence, if the authors want to start with an ill-posed equation it should be mentioned at the start. At least the Langevin dynamics considered should be explicitly mentioned in the main text. Since this work claims to be relevant to biological systems, it is also of significance to highlight the motivation for using the ill-posed equation rather than a well-posed equation. The authors refer to the non-equilibrium nature of the dynamics but it is not mentioned what non-equilibrium dynamics to authors have in mind. To properly analyze an overdamped Langevin dynamics a clear source of integrated timescales must be provided. As an example, one can write the dynamics as Eq. (1) \dot x = f(x) + g(x) \eta , which is ill-defined if the noise \eta is delta correlated in time but well-defined when \eta is exponentially correlated in time. One can of course look at the limit in which the exponential correlation goes to a delta correlation which leads to Eq. (1) interpreted in Stratonovich convention. The choice to use the Ito convention for Eq. (1) in this case is not justified.

We thank the reviewer for detailing their concerns with our model’s assumptions. We have addressed them in the common rebuttal.

(b) Generally, the manuscript talks of viscosity gradient but the equations deal with diffusion which is a combination of viscosity, temperature, particle size, and particle-medium interaction. There is no clear motivation provided for focus on viscosity (cytoplasm as such is a complex fluid) instead of just saying position-dependent diffusion constant. Maybe authors should use viscosity only when talking of a context where the existence of a viscosity gradient is established either in a real experiment or in a thought experiment.

The manuscript has been amended to use only “diffusivity” to avoid confusion.

(c) The section "Viscophoresis drives particle accumulation" seems to not have new results. Fig. 1 verifies the numerical code used to obtain the results in the later sections. If that is the case maybe this section can be moved to supplementary or at least it should be clearly stated that this is to establish the correctness of the simulation method. It would also be nice to comment a bit more on the choice of simulation methods with changing hopping sizes instead of, for example, numerically solving stochastic ODE.

The main point of this section and of Fig. 1 is the diffusive lensing effect itself: the accumulation of particles in lower-diffusivity areas. To the best of our knowledge, diffusive lensing has not been reported elsewhere as a specific outcome of non-isothermal interpretations of diffusion, with potential relevance to nonequilibrium subcellular motilities. The simulation method has been fully described in the Methods section, and the code has also been shared (see Code Availability).

A minor comment, the statement "the physically appropriate convention to use depends upon microscopic parameters and timescale hierarchies not captured in a coarse-grained model of diffusion." is not true as is noted in the references that authors mention, a correct coarse-grained model provides a suitable convention (see also Phys. Rev. E, 70(3), 036120., Phys. Rev. E, 100(6), 062602.).

This has been addressed in the common rebuttal.

(d) The section "Interaction-mediated clustering is affected by viscophoresis" makes an interesting statement about the positioning of clusters by a viscous gradient. As a theoretical calculation, the interplay between position-dependent diffusivity and phase separation is indeed interesting, but the problem needs more analysis than that offered in this manuscript. Just a plot showing clustering with and without a gradient of diffusion does not give enough insight into the interplay between density-dependent diffusion and position-dependent diffusion. A phase plot that somehow shows the relative contribution of the two effects would have been nice. Also, it should be emphasized in the main text that the inter-particle interaction is through a density-dependent diffusion constant and not a conservative coupling by an interaction potential.

The density-dependence has been added from the Methods to the main text. The goal of the work is to present lensing as a natural outcome of the parameter choices we make and present its effects as they relate to clustering and commonly used biophysical methods to probe dynamics within cells. A dense sampling of the phase space and how it is altered as a function of diffusivity, and the subsequent interpretation, lie beyond the scope of the present work but offer exciting future directions of study.

(e) The section "In silico microrheology shows that viscophoresis manifests as anomalous diffusion" the authors show that the MSD with and without spatial heterogeneity is different. This is not a surprise - as the underlying equations are different the MSD should be different.

The goal here is to compare and contrast the ways in which homogeneous and heterogeneous diffusion manifest in simulated microrheology measurements. We hope that an altered saturation MSD, as is observed in our simulations, provokes interest in considering lensing while modeling experimental data.

There are various analogies drawn in this section without any justification:(i) "the saturation MSD was higher than what was seen in the homogeneous diffusion scenario possibly due to particles robustly populating the bulk milieu followed by directed motion into the viscous zone (similar to that of a Brownian ratchet, (Peskin et al., 1993))."

In case of (i), the Brownian ratchet is invoked as a model to explain directed accumulation. We have removed this analogy to avoid confusion as it is not delved into further over the course of our work.

(ii) "Note that lensing may cause particle displacements to deviate from a Gaussian distribution, which could explain anomalous behaviors observed both in our simulations and in experiments in cells (Parry et al., 2014)." Since the full trajectory of the particles is available, it can be analyzed to check if this is indeed the case.

This has been addressed in the common rebuttal.

(f) The final section "In silico FRAP in a heterogeneously viscous environment ... " studies the MSD of the particles in a medium with heterogeneous viscous patches which I find the most novel section of the work. As with the section on inter-particle interaction, this needs further analysis.

We thank the reviewer for their appreciation. In presenting these three sections discussing the effects of diffusive lensing, we intend to broadly outline the scope of this phenomenon in influencing a range of behaviors. Exploring the directions further comprise promising future directions of study that lie beyond the scope of this manuscript.

To summarise, as this is a theory paper, just showing MSD or in silico FRAP data is not sufficient. Unlike experiments where one is trying to understand the systems, here one has full access to the dynamics either analytically or in simulation. So just stating that the MSD in heterogeneous and homogeneous environments are not the same is not sufficient. With further analysis, this work can be of theoretical interest. Finally, just as a matter of personal taste, I am not in favor of the analogy with optical lensing. I don't see the connection.

We value the reviewer’s interest in investigating the causes underlying the differences in the MSDs and agree that it represents a promising future area of study. The main point of this section of the manuscript was to make a connection to experimentally measurable quantities.

**Reviewer #2 (Public Review):**
Summary:The authors study through theory and simulations the diffusion of microscopic particles and aim to account for the effects of inhomogeneous viscosity and diffusion - in particular regarding the intracellular environment. They propose a mechanism, termed "Diffusive lensing", by which particles are attracted towards high-viscosity regions where they remain trapped. To obtain these results, the authors rely on agent-based simulations using custom rules performed with the Ito stochastic calculus convention, without spurious drift. They acknowledge the fact that this convention does not describe equilibrium systems, and that their results would not hold at equilibrium - and discard these facts by invoking the fact that cells are out-of-equilibrium. Finally, they show some applications of their findings, in particular enhanced clustering in the high-viscosity regions. The authors conclude that as inhomogeneous diffusion is ubiquitous in life, so must their mechanism be, and hence it must be important.Strengths:The article is well-written, and clearly intelligible, its hypotheses are stated relatively clearly and the models and mathematical derivations are compatible with these hypotheses.

We thank the reviewer for their appreciation.

Weaknesses:The main problem of the paper is these hypotheses. Indeed, it all relies on the Ito interpretation of the stochastic integrals. Stochastic conventions are a notoriously tricky business, but they are both mathematically and physically well-understood and do not result in any "dilemma" [some citations in the article, such as (Lau and Lubensky) and (Volpe and Wehr), make an unambiguous resolution of these]. Conventions are not an intrinsic, fixed property of a system, but a choice of writing; however, whenever going from one to another, one must include a "spurious drift" that compensates for the effect of this change - a mathematical subtlety that is entirely omitted in the article: if the drift is zero in one convention, it will thus be non-zero in another in the presence of diffusive gradients. It is well established that for equilibrium systems obeying fluctuation-dissipation, the spurious drift vanishes in the anti-Ito stochastic convention (which is not "anticipatory", contrarily to claims in the article, are the "steps" are local and infinitesimal). This ensures that the diffusion gradients do not induce currents and probability gradients, and thus that the steady-state PDF is the Gibbs measure. This equilibrium case should be seen as the default: a thermal system NOT obeying this law should warrant a strong justification (for instance in the Volpe and Wehr review this can occur through memory effects in robotic dynamics, or through strong fluctuation-dissipation breakdown). In near-equilibrium thermal systems such as the intracellular medium (where, although out-of-equilibrium, temperature remains a relevant and mostly homogeneous quantity), deviations from this behavior must be physically justified and go to zero when going towards equilibrium.

Considering that the physical phenomena underlying diffusion span a range of timescales (particle relaxation, noise, environmental correlation, et cetera), we disagree with the assertion that all types of subcellular diffusion processes can be modeled as occurring at thermal equilibrium: for example, one can easily imagine memory effects arising in the presence of an appropriate hierarchy of timescales. We have added references that describe in more detail the way in which the comparison of timescales can dictate the applicability of different conventions. We also refer the referee to the common rebuttal section of our response in which we discuss factors that govern the choice of the interpretation. The adiabatic elimination arguments highlighted in (Kupferman et al. 2004) provide a clear description of how relevant particle and environment-related timescales can inform the choice of stochastic calculus to use.

With regards to the use of the term “anticipatory” to refer to the isothermal interpretation, we refer to the comment in (Volpe and Wehr 2016) of the Itô interpretation “not looking into the future”. In any case, whether anticipatory or otherwise, the interpretation’s effect on our model remains unchanged, as highlighted in the section in the Appendix on the conversion between different conventions; this section has been added to minimize confusion about the effects of the choice of convention on lensing.

Here, drifts are arbitrarily set to zero in the Ito convention (the exact opposite of the equilibrium anti-Ito), which is the equilibrium equivalent to adding a force (with drift $- grad D$) exactly compensating the spurious drift. If we were to interpret this as a breakdown of detailed balance with inhomogeneous temperature, the "hot" region would be effectively at 4x higher temperature than the cold region (i.e. 1200K) in Fig 1A.

Our work is based on existing observations of space-dependent diffusivity in cells (Garner et al., 2023; Huang et al., 2021; Parry et al., 2014; Śmigiel et al., 2022; Xiang et al., 2020). These papers support a definitive model for the existence of space-dependent diffusivity without invoking space-dependent temperature.

It is the effects of this arbitrary force (exactly compensating the Ito spurious drift) that are studied in the article. The fact that it results in probability gradients is trivial once formulated this way (and in no way is this new - many of the references, for instance, Volpe and Wehr, mention this).

Addressed in the common rebuttal.

Enhanced clustering is also a trivial effect of this probability gradient (the local concentration is increased by this force field, so phase separation can occur). As a side note the "neighbor sensing" scheme to describe interactions is very peculiar and not physically motivated - it violates stochastic thermodynamics laws too, as the detailed balance is apparently not respected.The neighbor-sensing scheme used here is just one possible model of an effective attractive potential between particles. Other models that lead to density-dependent attraction between particles should also provide qualitatively similar results as ours; this offers an interesting prospect for future research.Finally, the "anomalous diffusion" discussion is at odds with what the literature on this subject considers anomalous (the exponent does not appear anomalous).

This has been addressed in the common rebuttal, and the relevant part of the manuscript has been modified to avoid confusion.

The authors make no further justification of their choice of convention than the fact that cells are out-of-equilibrium, leaving the feeling that this is a detail. They make mentions of systems (eg glycogen, prebiotic environment) for which (near-)equilibrium physics should mostly prevail, and of fluctuation-dissipation ("Diffusivity varies inversely with viscosity", in the introduction). Yet the "phenomenon" they discuss is entirely reliant on an undiscussed mechanism by which these assumptions would be completely violated (the citations they make for this - Gnesotto '18 and Phillips '12 - are simply discussions of the fact that cells are out-of-equilibrium, not on any consequences on the convention).Finally, while inhomogeneous diffusion is ubiquitous, the strength of this effect in realistic conditions is not discussed (this would be a significant problem if the effect were real, which it isn't). Gravitational attraction is also an ubiquitous effect, but it is not important for intracellular compartmentalization.

The manuscript text has been supplemented with additional references that detail the ways in which the comparison of timescales can dictate how one can apply different conventions. We refer the reviewer to the common rebuttal section of our response where we detail factors that dictate the choice of the convention to use. As previously noted, the adiabatic elimination arguments highlighted in (Kupferman et al., 2004) provide a prescription for how different timescales are to be considered in deciding the choice of stochastic calculus to use.

With regards to the strength of space-dependent diffusivity in subcellular milieu, various measurements of heterogeneous diffusivity have been made both across different model systems and via different modalities, as cited in our manuscript. (Garner et al. 2023) used single-particle tracking to determine over 100-fold variability in diffusivity within individual S. pombe cells. Single-molecule measurements in (Xiang et al. 2020) and (Śmigiel et al. 2022) reveal an order-of-magnitude variation in tracer diffusion in mammalian cells and multi-fold variation in *E. coli* cytoplasm respectively. Fluorescence correlation spectroscopy measurements in (Huang et al. 2022) have found a two-fold increase in short-range diffusion of protein-sized tracers in X. laevis extracts. We have also added a reference to a study that uses 3D single particle tracking in the cytosol of a multinucleate fungus, A. gossypii, to identify regions of low-diffusivity near nuclei and hyphal tips (McLaughlin et al. 2020). Many of these references deploy particle tracking and investigate how mesoscale-sized particles (i.e. tracers spanning biologically relevant size scales) are directly impacted by space-dependent diffusivity. Therefore, we base our model on not only space-dependent diffusivity being a well-recognized feature of the cellular interior, but also on these observations pertaining to mesoscale-sized particles’ motion along relevant timescales.

These measurements are also relevant to the reviewer’s question about the strength of the effect, which depends directly on the variability in diffusivity: for ten- or a hundred-fold diffusivity variations, the effect would be expected to be significant. In case of using the Itô convention directly, the contrast in concentration gradient is, in fact, that of the diffusivity gradient.

To conclude, the "diffusive lensing" effect presented here is not a deep physical discovery, but a well-known effect of sticking to the wrong stochastic convention.

As detailed in the various responses above, we respectfully disagree with the notion that there exists a singular correct stochastic convention that is applicable for all cases of subcellular heterogeneous diffusion. Further, as detailed in Volpe and Wehr 2016 and as detailed in the Appendix, it is possible to convert between conventions and that an isothermal-abiding stochastic differential equation may be suitably altered, by means of adding a drift term, to an Itô-abiding stochastic differential equation; therefore, one can observe diffusive lensing without discarding the isothermal convention if the latter were modified. Indeed, it is only the driftless (or canonical) isothermal convention that does not allow for diffusive lensing.